# A Flexible and Highly Sensitive Pressure Sensor Based on AgNWs/NRLF for Hand Motion Monitoring

**DOI:** 10.3390/nano9070945

**Published:** 2019-06-29

**Authors:** Yi Sun, Zhaoqun Du

**Affiliations:** 1Key Laboratory of Textile Science & Technology (Donghua University), Ministry of Education, College of Textiles, Donghua University, Shanghai 201620, China; 2Jiangxi Provincial Center for Quality Inspection and Supervision on Down Products, Gongqingcheng 332020, China

**Keywords:** pressure sensor, silver nanowires, natural rubber latex foam, piezoresistive, hydrophobicity

## Abstract

Flexible, highly sensitive, easy fabricating process, low-cost pressure sensors are the trend for flexible electronic devices. Inspired by the softness, comfortable, environmental friendliness and harmless of natural latex mattress, herein, we report an agile approach of constructing a flexible 3D-architectured conductive network by dip-coating silver nanowires (AgNWs) on the natural rubber latex foam (NRLF) substrate that provide the 3D micro-network structure as the skeleton. The variation of the contact transformed into the electrical signal among the conductive three-dimensional random networks during compressive deformation is the piezoresistive effect of AgNWs/NRLF pressure sensors. The resulting AgNWs/NRLF pressure sensors exhibit desirable electrical conductivity (0.45–0.50 S/m), excellent flexibility (58.57 kPa at 80% strain), good hydrophobicity (~128° at 5th dip-coated times) and outstanding repeatability. The AgNWs/NRLF sensors can be assembled on a glove to detect hand motion sensitively such as bending, touching and holding, show potential application such as artificial skin, human prostheses and health monitoring in multifunctional pressure sensors.

## 1. Introduction

Flexible sensors [1,2] are electronic systems that use sensor technology combined with new material technology to mimic protection, perception and regulation of human skin [3,4,5,6,7]. The development of flexible sensors mainly focuses on flexibility, low detection limit, high sensitivity, low cost, etc. Flexible sensors have different functions for different problems that have been applied to many aspects of human life [8,9,10], including human prostheses, pressure imaging, health monitoring, clinical medicine, industrial life, human–computer interaction and other fields. According to the sensing mechanisms, flexible sensors mainly include transistor sensors [11,12,13], capacitive sensors [14,15,16], piezoelectric sensors [17,18] and piezoresistive sensors [19].

The developments of portable electrical devices [20], have increased the demand for flexible sensors [21,22], for example artificial skin [23,24,25]. Recently, several highly sensitive flexible pressure sensors [26] have been manufactured and some pressure sensors have been fabricated based on the piezoresistive sensing mechanism [27,28]. The resistance-strain effect refers to the change in electrical resistance of materials when it is deformed. According to the resistance-strain effect, sensors based on various conductive materials are designed by measuring the change in resistance when the objects are deformed. An efficient method is to combine composite conductive networks to piezoresistive sensors. Among the large number of conductive materials, silver nanowires were more convenient to prepare, have high conductivity and could form natural conductive networks structure. Silver nanowires (AgNWs) have widespread concern due to their excellent electrical and mechanical properties in sensors [29]. Over the past few years, the polyol process has been the most promising method for preparing AgNWs [30,31,32], however, few studies attempted to research the effect of the stirring rate of the synthesize process. In order to obtain the length of AgNWs that meet our needs, we did experiments to determine the optimum stirring rate in this study. Recently, most conductive applications of AgNWs are currently two-dimensional networks, and different coating methods were utilized for AgNWs on different substrates. Electrophoretic deposition, spin coating, and spray coating were used for AgNWs fabrication with flexible films [33,34,35] Also, AgNWs were performed by an inkjet printing method on the substrate of coated PET and photo paper [36]. In this study, we used the dip-coating method to achieve the natural and random distribution of silver nanowires without destroying the length of AgNWs.

Sensitivity and compressibility should be considered when manufacturing the excellent performance sensors, that the response of the sensor to the external stimulation like compression is primarily dependent on changes in electrical characteristics such as resistance. In recent years, because of the excellent electrical conductivity and compressibility, three-dimensional (3D) structures reinforced flexible sensors are the hot spot of research. The combination of silver nanowires and Polydimethylsiloxane (PDMS) is more common in sensor manufacturing, PDMS substrate coated with AgNWs was used as the top electrode, and polyvinylidene fluoride (PVDF), have the excellent performance with the dielectric layer of high sensitivity and low detection limit [37]. The silver nanowire networks, combined with a sandwich structure of PDMS elastomer, exhibits strong piezoresistivity that the gauge factors is in 2 to 14 and high tensile properties up to 70% [38]. Several flexible sensors have been fabricated, such as conductive networks or films, as well as other conductive foams or sponges [39] embedded in different substrates in purpose of replacing the conventional rigid sensors. Piezoresistive sensors were developed based on the commercial elastic foams and traditional porous structures such as polyurethane (PU) sponges with high flexibility and elasticity [40], to build conductive networks such as AgNWs on their skeletons. Porous polymers or sponges that were used as conductive layers have unique advantages, such as good mechanical properties, natural network structure, inexpensive host matrices, and being easy to manufacture [41]. However, most of the substrates used currently are PU sponges, which have high chemical composition [42]. At present, there is little research on the combination of silver nanowires and natural latex in the field of sensing, and for natural environmental protection and excellent elasticity, natural latex porous materials are chosen.

In this paper, AgNWs were immersed into a matrix of the natural rubber latex foam (NRLF) to fabricate a 3D structured and piezoresistive AgNWs/NRLF pressure sensor with high linearity and sensitivity. Pressure sensors of AgNWs/NRLFs had good electrical and mechanical performance due to the 3D interconnection of the NRLF skeleton coated with AgNWs [43]. The synthesis method and the factors affecting the length and fineness of silver nanowires are discussed. The electrical responsive under cyclic loading of compression and the hydrophobicity of AgNWs/NRLF sensors are investigated. A hand glove integrated with AgNWs/NRLF sensors on each five fingers is developed for the detection of hand motion. 

## 2. Experimental

### 2.1. Materials

The following materials were used in this study: polyvinylpyrro-lidone (PVP K-30) (*M*_W_ ≈ 40,000–55,000), silver nitrate (AgNO_3_) (≥99.8%) were purchased from Sinopharm Chemical Reagent Co., Ltd. (Shanghai, China). Ethylene glycol (EG) (≥99.0%), sodium chloride (NaCl) (≥99.5%), acetone (≥99.5%), ethanol (≥95.0%) were purchased from Shanghai Lingfeng Chemical Reagent Co., Ltd. (Shanghai, China). Natural rubber latex foams (NRLFs) were purchased from mattress market. All chemical reagents were of analytical purity and could be used without further purification.

### 2.2. Preparation of AgNWs

Silver nanowires (AgNWs) were synthesized by the common polyol process [43]. Ethylene glycol, which is a common polyol solution, has been widely used to reduce Ag^+^ during the synthesis of silver nanowires. All glassware used in the experiments were washed with deionized water. The schematic of the synthesis procedure is shown in Figure 1.

In a typical Polyol synthesis route, the 30 mL EG was heated and thermally stabilized at 170 °C for 1 h with a magnetic stirrer (stirring speed = 260 rpm), 170 °C is a sufficient temperature to evaporate water and enhance the reducing ability of EG. After the EG solution was cooled down to the room temperature, 0.4g PVP, 0.2 g AgNO_3_ and 20 mL heated EG were stirred in a flask for 30 min under shading treatment. Afterward, 200 μL of 0.1 M NaCl in EG solution was added dropwise to the above solution at a rate of 0.5 mL/min. After injecting all NaCl solution, the solution was heated at 170 °C for 2 h for the complete formation of AgNWs. After cooling the resulting solution, a large amount of acetone was added to the solution at a ratio of 5:1. The mixture was washed by centrifugal every 10 min at 4500-5000 rpm two times, then washed with ethanol every 5 min at 5000 rpm three times to remove the excess PVP, EG and other impurities. After the impurity removal step, AgNWs were stored in ethanol for further experiments.

### 2.3. Preparation of AgNWs/NRLFs

Natural latex is mainly composed of natural materials, and only undergoes post-processing. Porous natural rubber latex foams (NRLFs) were cut into pieces in the form of a square with 15mm sides and a thickness of 5mm for resistance and compressive tests. Width by length of 6 × 40 mm^2^ for hydrophobic tests. All pieces were washed three times with deionized water, immersed in ethanol for 1 h, and then dried at 60 °C in the middle of the oven for 2 h. After that, the pieces were immersed into the AgNWs of ethanol solution (~4 mg/mL) for one day, dried at 60 °C for 2 h. By repeating the dip-coating step five times with each time interval being 24 h, eventually AgNWs/NRLFs were formed with different conductivity (marked as 1th, 2th, 3th, 4th, and 5th dip-coated conductivity). After drying the AgNWs/NRLF and evaporating the ethanol, the color of the NRLF changed from white to gray (Figure 2a). Figure 2b showed the fabrication of AgNWs/NRLF sensors, the Copper wires were adhered to the top and bottom of the AgNWs/NRLF to test the electrical properties during compression.

### 2.4. Characterization

Surface morphologies of the silver nanowires and dip-coated NRLFs were examined with a scanning electron microscope (SEM, Flex) under 15 kV. All the samples were sputtered with gold before testing.

A compression machine (Nantong Hongda Experiment Instruments Co., Ltd., Nantong, China) was employed for compressive test. The force and displacement were obtained to analyze the mechanical properties, where the initial height of compression was 7 mm and the presser foot area was 38 mm^2^. The strain amplitude changed from 60% to 80% with the strain rate, and recovery rate was 10 mm/min.

The electrical resistance of the AgNWs/NRLF was measured using a two-wire resistance technique, through Keithley 2701 data collector (Tektronix, Beaverton, OR, USA) connected to the laptop for data acquisition. The AgNWs/NRLF was sandwiched between two copper sheets which work as electrodes. Two copper wires were attached to the electrodes respectively and connected to the measuring unit. Then, the electrical properties were measured using the previously described instrument under different conditions of pressure to investigate the piezoresistive response and performances of fabricated conductive materials as pressure sensors [44].

The hydrophobicity of AgNWs/NRLFs were estimated of the contact angles (CA) of the liquid drops measured by Optical Contact Angle Meter OCA15EC (Dataphysics, Germany). The test used a hanging drop method where droplets (5 μL) could come into contact with the surface of the samples when leaving the needle. At least three readings per sample were recorded.

## 3. Results and Discussion

### 3.1. Microstructure of AgNWs/NRLFs

In the process of the whole test, silver nanowires cannot be synthesized without the presence of PVP. The synthesized solution needs to be washed by centrifugation to remove the remaining PVP of the chemical reaction. The growth principle of silver nanowires is that the most active surface of nano-silver cubes is (111), while silver nanoparticles preferentially grow along the (111) direction. During the growth of silver nanocrystal, PVP acts as a tubular tunnel to conduct the crystal faces. Due to the restraint-force of PVP, the silver nanoseeds first are covered with PVP shells, silver seeds grow on the active face (111) of the highest surface a higher speed, eventually the silver nanowires are obtained [45]. A temperature of 170 °C was selected during the synthesis to synthesize longer nanowires, which resulted in a wider diameter distribution of AgNWs. The final mixed solution is yellow suspension and this result is confirmed and displayed in the illustration in Figure 3a. Figure 3a,b demonstrated that uniform AgNWs were achieved, the average length of the AgNWs was 20 ± 5 μm and the average diameter was found to be 88 ± 6 nm. Figure 3c,d showed the NRLF dip-coated one time with AgNWs solution under different magnifications. The results showed that AgNWs have excellent adhesion with micro-scale fibers because of entanglement and electrostatic attraction, and could attach well to the NRLF and form continuous and evenly distributed network structure on the surface of NRLF during the dip-coating process.

In order to synthesize longer nanowires using the polyol process [46], the main possible factors that influence the results are temperature, stirring rate, and ratio of AgNO_3_ to PVP. These mainly study the effect of the stirring rate on synthesis. In the development of the optimal experimental solution, in order to obtain silver nanowires with moderate length and diameter, try to use magnetic stirring during the synthesis but the result was not ideal (Appendix A). The AgNWs were carefully chosen for its high conductivity and the ability to form continuous and well distributed nano-network structure combined with NRLF.

### 3.2. Hydrophobicity of AgNWs/NRLFs

The wettability of a solid surface was determined by the chemical composition of the solid surface and its three-dimensional microstructure [47]. Two ways were usually used to improve the contact angle and hydrophobicity: (1) modify chemical composition of the surface chemically to reduce its free energy; (2) change the three-dimensional microstructure and improve the roughness of the solid surface [48,49]. The experiment of silver nanowires dip-coating increased the conductivity of the NRLFs and form the random three-dimensional structure due to the adsorption of the silver nanowires as the above way (2). Close-up views of optical microscope images Figure 4a showed that NRLFs have properties including porous, interconnected and three-dimensional structures. Figure 4b showed the surfaces of AgNWs/NRLFs formed many tiny structures similar to the ladybug shells and arranged closely. This special structure of micro-nano-scale roughness can greatly improve the contact angle of the water drop on it. When a large drop was added to the surface of AgNWs/NRLFs, the contact angle indicated high hydrophobicity (>120–130°) with dip-coated treatment (Figure 4c). The water contact angle (CA) of AgNWs/NRLFs after five dip-coating cycles could achieve 128° (Figure 4d), but when it was dip-coated after six cycles, its water contact angle (CA) decreased as 105° (Appendix A), the AgNWs/NRLFs changed from hydrophobic to hydrophilic. The results showed the AgNWs/NRLFs had certain hydrophobicity and could be used stably in the air with humidity.

### 3.3. Mechanical Properties of AgNWs/NRLFs

The mechanical properties of sensors are important to the electrical performance [50], such as the compressibility of the piezoresistive sensor [51,52] is directly related to its resistance response when subjected to the external stress aimed at indirectly assessing their sensitivity of piezoresistive behavior after AgNWs modification. The matrix of the sensor is natural latex, which has been favored by consumers because of its softness and durability. Here, the mechanical properties of pretreatment and dip-coated treatment on the mechanical properties of samples were further discussed and the NRLF and AgNWs/NRLF were tested at a compression rate of 10 mm/min under different compressive strain amplitude. Figure 5a showed the main working structure of the compressive system included the upper platen and bottom platen, and the thickness of the sample was 5 mm. As shown in Figure 5b,c, the compressive process of NRLF and AgNWs/NRLF were observed under 60% and 80% compressive strain. The stress of NRLF was about 8.45 kPa at 60% strain and 17.41 kPa at 80% strain; the stress of the NRLF increased slowly before 60% strain and increased quickly and tend to be linear after 60% strain; and each stress could almost return to the initial point for each compressive strain without loading (Figure 5d). As shown in Figure 5e, the form of stress changed with the strain of AgNWs/NRLF was similar to NRLF but the final stress was 58.57 kPa at 80% strain which increased compared with NRLF, reflecting the coating modification would reduce the elasticity due to the stiffness caused by silver nanowires.

All of these proved the excellent elasticity and recovery of NRLF. The influence on the mechanical property of coating modification was also investigated by compressing the conductive AgNWs/NRLF. As seen above, the results showed that the original NRLF and the dip-coated AgNWs/NRLF could be used as flexible materials because of their good mechanical properties [53], so that the natural latex was used as a matrix of the pressure sensor [54] in this study.

### 3.4. Piezoresistive Performance of AgNWs/NRLFs

The electromechanical properties and the responsiveness of the AgNWs/NRLFs were studied and tested under cyclic compressive loading. The fifth sensors were chosen to characterize the electrical properties under different compressive strains due to the suitable compressive strength and stable sensing properties [55]. Here the conductivity was defined as σ=l/RS in which R was the resistance after dip-coating cycles, l was the length of AgNWs/NRLFs and S was the area of AgNWs/NRLFs. The length of the AgNWs/NRLFs were 15 mm, and the area were 225 mm^2^. The mass content of AgNWs on the NRLFs keep increasing with the dip-coating cycles so that more effective and conductive paths were formed to improve the conductivity. As shown in Figure 6, the final resistance of the 5th AgNWs/NRLFs were 140–160 Ω, and the conductivity were about 0.45–0.50 S/m. Figure 6 also showed the conductivity of 3rd dip-coated samples increased significantly and still increased in 4th and tended to be stable of 5th dip-coated samples. When silver nano-networks were not under pressure, the resistance returned to its original state.

$\Delta R/R_{0}=\left(R_{0}-R\right)/R_{0}$ represented relative resistance variation, in which $R_{0}$ was the initial resistance, R was the resistance during the test and ΔR was the change of resistance. As shown in Figure 7a, the responsiveness of 5th dip-coated AgNWs/NRLF sensors under finger pressure in compression and release cycles was tested. In each cycle, the finger applied pressure to the center of the AgNWs/NRLF sensors until the AgNWs/NRLF compressed to its limit and the resistance was about 89.9%. Besides, it was observed that the compressive stress of AgNWs/NRLF sensors could restore to the original state even after several cycles because the value of $\Delta R/R_{0}$ could return to 0 in each cycle. The LED was connected to AgNWs/NRLF with 1 V constant voltage power in the circuit that the LED was lit under the micro voltage with pressure, but was off without pressure (insert of Figure 7a). As shown in Figure 7b, the resistance of AgNWs/NRLF sensors was about 0, 63.8, 45.3 and 36.1% toward 0, 10, 20, and 50 g which as small deformation of the different framer weights by lifting and lowering framers at an interval of 3 s, and could almost return the initial value without loading. The R decreased and ΔR increased significantly, as more cross-link points were formed of AgNWs nano-networks inside the AgNWs/NRLF. Figure 7c,d showed the resistance was 50.7, 78.5 and 96.2% toward 20, 40 and 80% strain which as the large deformation in 150 cycles. As shown in Figure 7e, the electrical performance of AgNWs/NRLFs depend on the molecular chains in NRLFs matrix which has inherent superior elasticity and softness, and the silver nanowires network that could deform under the imposed stimulation.

All these resistance variations indicated the AgNWs/NRLF sensors could recover to the initial state in small and large deformation, and also exhibited the excellent piezoresistive effect and stable repeatability. The dip-coated samples formed continuous and evenly distributed network structure of the high conductive AgNWs and the 3D nano-networks could achieve overall electrochemical stability [38]. When the uniformly distributed AgNWs nano-networks were subjected to pressure, the contact points and areas between the wires and wires increased, so that the conductive loop grew, the conductivity increased, and thus the resistance decreased [56,57]. This also indicated that the AgNWs/NRLFs as flexible sensors can satisfy the electrical application under compressive deformation.

### 3.5. Applications in Hand Motion Monitoring

Due to the good compressive and piezoresistive sensing properties [58], so the AgNWs/NRLF sensor was used as pressure sensor to detect the human motion [59,60] in this work. The five 5th dip-coated AgNWs/NRLF pressure sensors were integrated on the thumb, index, middle, ring, and little fingers of the human glove and the response of sensors were tested. The pressure sensors were put into each finger-cot of the glove to detect the compressive force of each fingers during the human contact with the object by hand. As shown in Figure 8a, When the relative resistance $\Delta R/R_{0}$ of the middle finger increased to 9.1% as the middle finger was bent, and the $R/R_{0}$ of other fingers increased slightly ranged from 0 to 1.47% because other fingers were not static completely with the bending of the middle finger. As shown in Figure 8b, the $\Delta R/R_{0}$ of thumb and index fingers increased to 23.8 and 35.6% and other fingers almost be 0, except the middle finger had the same rule of the thumb and index fingers, and hence could monitor the motion of the hand [61,62]. Also, Figure 8c,d showed the results with different motions. Holding an egg or an apple indicated high sensitivity and significant variations of relative resistance. The results showed that the AgNWs/NRLF sensors located on the different fingers had excellent stable and repeatability that could detect the motion when the hand touched the external objects, which would be used in medical and health monitoring with broad application [63,64].

## 4. Conclusions

In summary, we showed a biomass-based pressure sensor made up of natural rubber latex foam (NRLF) combined with AgNWs by a simple dip-coating process. The layered structures of natural latex make it have ideal skeletons, which is the key to forming conductive networks, and the micro-network structure as the skeleton for supporting AgNWs networks. The pressure sensor developed with AgNWs dip-coated five times based on the NRLF, which confirmed to have high sensitivity, electromechanical stability and repeatability under cyclic compressive loading [37]. The resulting AgNWs/NRLF pressure sensors exhibit high electrical conductivity (0.45–0.50 S/m), excellent flexibility (58.57 kPa at 80% strain) and good hydrophobicity (~128° at 5th dip-coated times). The pressure sensors based on conductive AgNWs benefited from the multi-phase porous skeleton and organic nodes in natural latex. In the application research of pressure sensors, the glove which integrated five sensors could monitor the human hand activities in real time. Convenient fabrication methods and low-cost raw materials have significant advantages in mass production and wide application. Hence, suitable mechanical performance, hydrophobicity and sensing properties make AgNWs/NRLF pressure sensors become prospective and environmental protection materials for future application in flexible, hydrophobic, compressive and conductive sensor devices.

## Figures and Tables

**Figure 1 nanomaterials-09-00945-f001:**
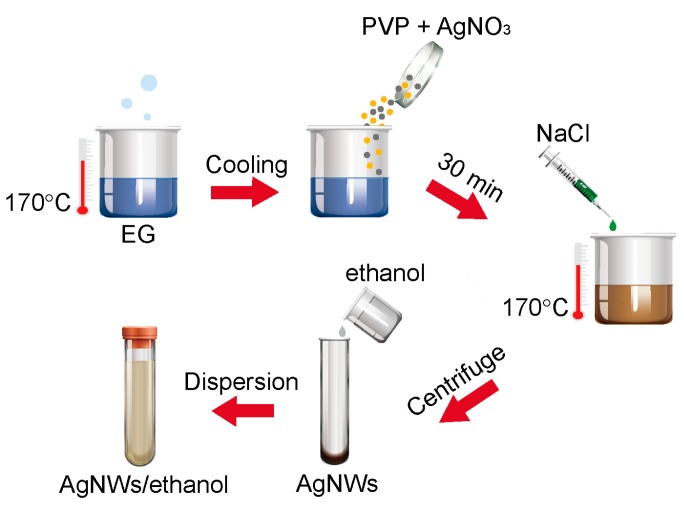
Schematic of the synthesis procedure of AgNWs. AgNW: silver nanowire.

**Figure 2 nanomaterials-09-00945-f002:**
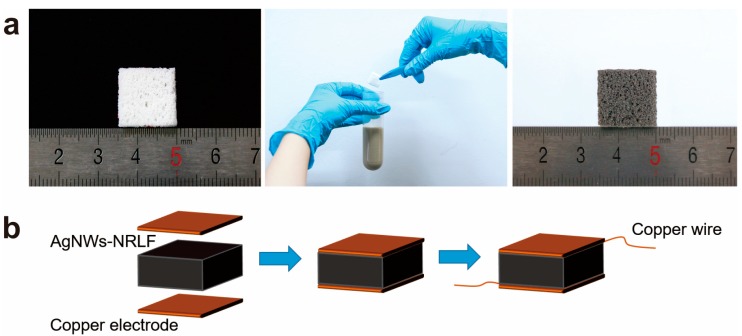
Preparation of AgNWs/NRLF pressure sensor. (**a**) Schematic illustration of the fabrication procedure of AgNWs/NRLF. (**b**) Schematic of the fabrication process of AgNWs/NRLF pressure sensor.

**Figure 3 nanomaterials-09-00945-f003:**
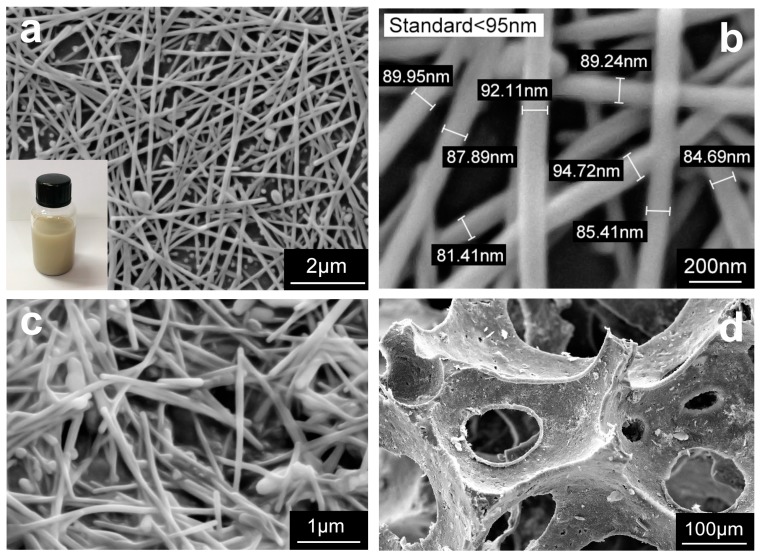
SEM micrographs of various parts of the samples: (**a**) SEM image for network structure of AgNWs at low stirring rate (inset: AgNWs suspensions in ethanol); (**b**) SEM confirms that the diameters of silver nanowires less than 100 nm; (**c**,**d**) SEM images of NRLF under different magnifications after AgNWs solution coating with one time.

**Figure 4 nanomaterials-09-00945-f004:**
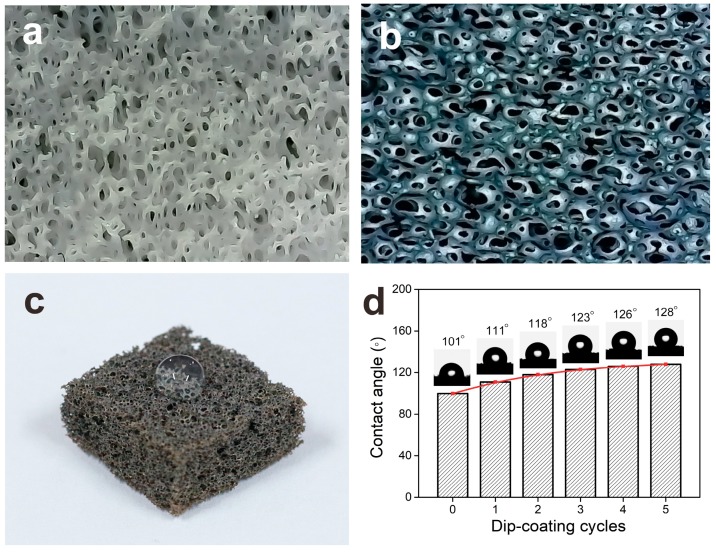
Optical images of (**a**) surface of the original NRLF. (**b**) hydrophobic surface of the AgNWs/NRLF. (**c**) large drop of deionized water on the AgNWs/NRLF. (**d**) change in contact angles with different dip-coating cycles.

**Figure 5 nanomaterials-09-00945-f005:**
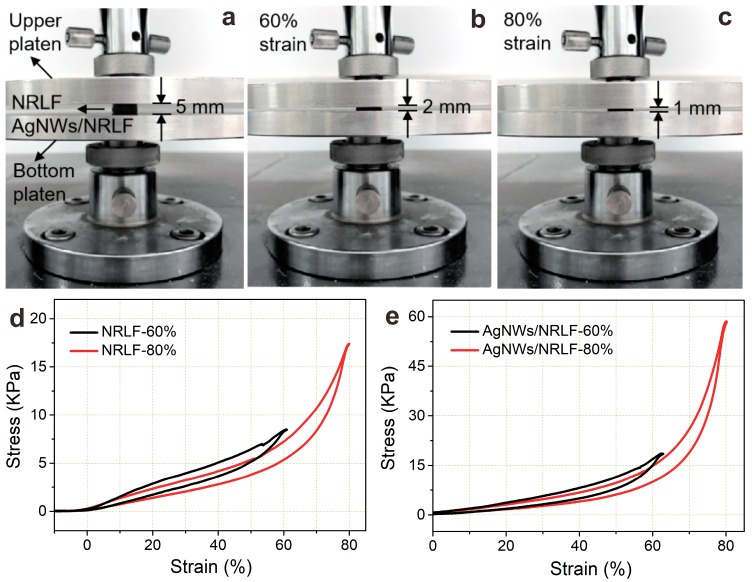
(**a**) the compressive testing system and NRLFs/AgNWs/NRLFs. (**b**,**c**) compressive strains of 60%, 80%. (**d**,**e**) compressive stress variation of NRLFs and AgNWs/NRLFs with different compressive strain amplitude.

**Figure 6 nanomaterials-09-00945-f006:**
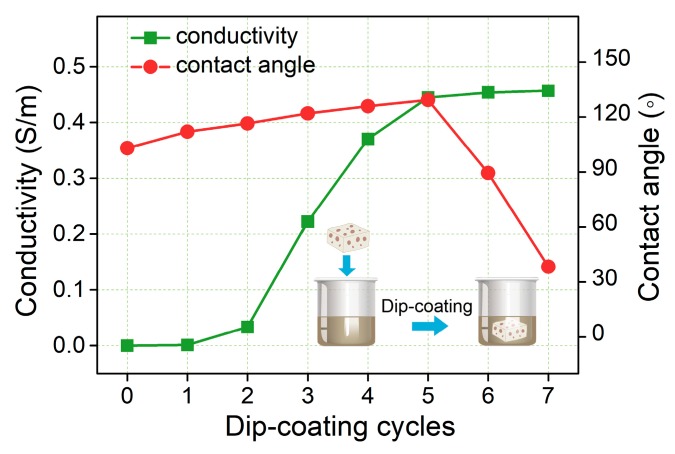
Conductivity and contact angle of AgNWs-NRLF with different dip-coating cycles.

**Figure 7 nanomaterials-09-00945-f007:**
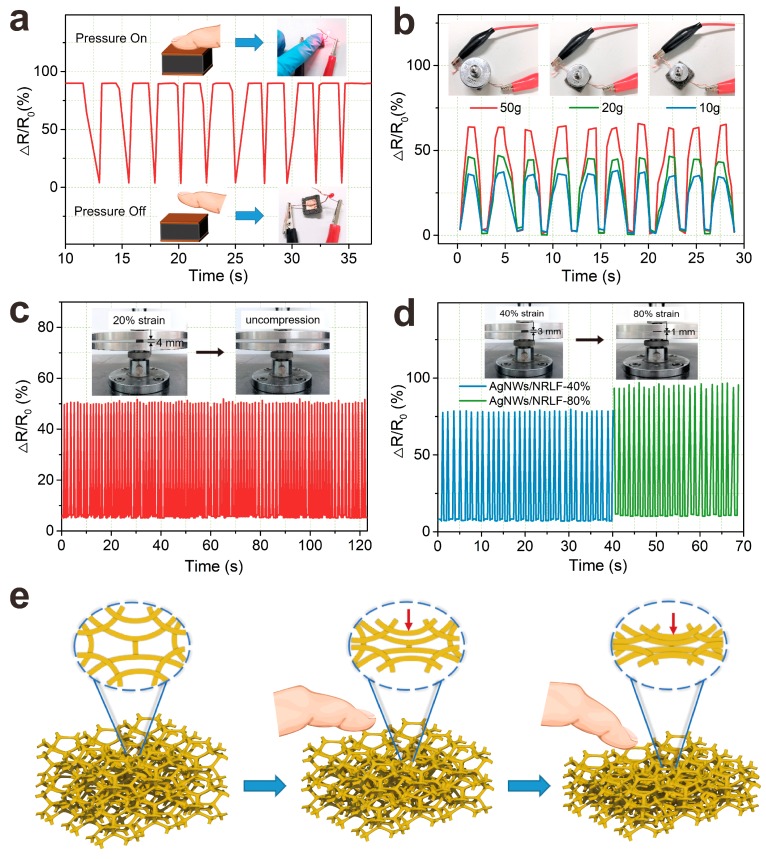
(**a**,**b**) relative resistance variation of the AgNWs/NRLF sensors under small compression strains (finger pressure and cyclic 50 g, 0 g, 10 g weight); (**c**,**d**) cyclic piezoresistive reaction of the AgNWs/NRLF sensors under large compression strains (20%, 40% and 80%); (**e**) schematic of piezoresistive effect under compressive deformation.

**Figure 8 nanomaterials-09-00945-f008:**
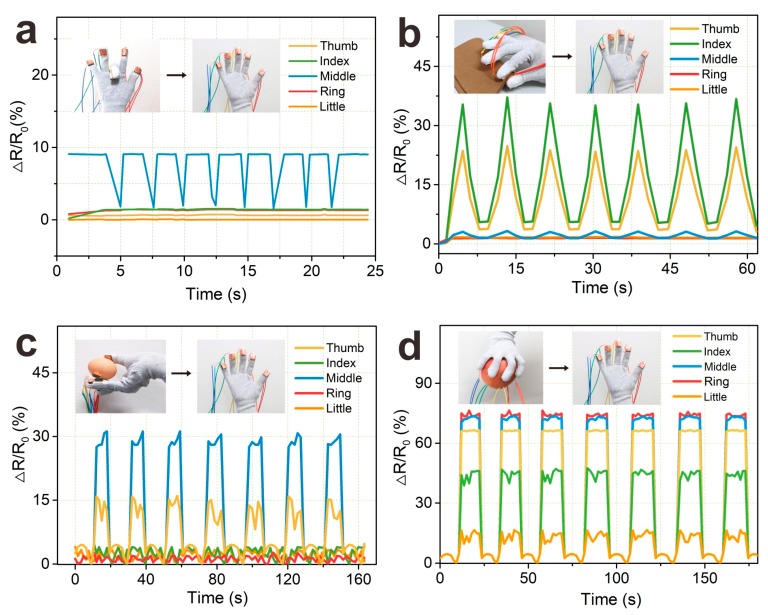
Relative resistance variation of AgNWs/NRLF under cyclically hand gestures, including (**a**) middle finger bending; (**b**) touching fabric; (**c**) holding egg; (**d**) holding apple.

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
