# Peer review of "A Flexible and Highly Sensitive Pressure Sensor Based on AgNWs/NRLF for Hand Motion Monitoring"

_nanomaterials, 2019, doi:10.3390/nano9070945_

Round 1
Reviewer 1 Report
The current manuscript reports on a biomass-based pressure sensor made up of natural rubber latex foam combined with AgNWs. The pressure sensor developed showed high sensitivity, electromechanical stability and repeatability under cyclic compressive loading. The results are clearly presented and supported by the technical information provided by the authors, the novelty of the results and the resulting added value for nanotechnology and materials science is not highlighted in the introduction section and is not obvious. Therefore, I suggest that the authors clearly highlight what is novel with respect to the state of the art and how their method advances the field of Ag NW based sensors before the manuscript is accepted for publication in this journal. Also, a short overview of the different coating/application methods utilized for Ag NWs on different substrates should be added in the introduction.
Other minor comments:
1. In the introduction and across the text, brackets, [], are mostly missing from the references. This should be corrected.
2. In the introduction section, L70, "...of silver nanowires were discussed", please change past tense into present or future tense.
3. In L139 "...silver nanowires cannot synthesize without the presence of PVP", please use passive voice instead i.e. "cannot be synthesized"
Reviewer 2 Report
This paper show a biomass-based pressure sensor using a simple dip process in natural rubber latex foam and making use of AgNWs the piezo-resistivity for sensing. It is interesting to read and I would like to recommend it for publication subject to minor corrections,
1. Please use a same theme for references that is either use [ ] brackets or not use them at all but please dont use some time [ ] and sometime no brackets.
2. Please use the full chemical name first time then say PDMS.
3. In a flexible sensing system, it would be nice if the authors could comments or show that the sensor are capable to sense the stretches of the glove’s skin. This means that if the sensors will be placed on the top parts of the fingers and the sensors will stretch when the finger will bend.
